# CLUE: NON-PARAMETRIC VERIFICATION FROM EXPERIENCE VIA HIDDEN-STATE CLUSTERING

## ABSTRACT

Assessing the quality of Large Language Model (LLM) outputs presents a critical challenge. Previous methods either rely on text-level information (e.g., reward models, majority voting), which can overfit to superficial cues, or on calibrated confidence from token probabilities, which woule fail on less-calibrated models. Yet both of these signals are, in fact, partial projections of a richer source of information: the model's internal hidden states. Early layers, closer to token embeddings, preserve semantic and lexical features that underpin text-based judgments, while later layers increasingly align with output logits, embedding confidence-related information. This paper explores hidden states directly as a unified foundation for verification. We show that the correctness of a solution is encoded as a geometrically separable signature within the trajectory of hidden activations. To validate this, we present **CLUE (Clustering and Experience-based Verification)**, a deliberately minimalist, non-parametric verifier. With no trainable parameters, CLUE only summarizes each reasoning trace by an hidden state delta and classifies correctness via nearest-centroid distance to "success" and "failure" clusters formed from past experience. The simplicity of this method highlights the strength of the underlying signal. Empirically, CLUE consistently outperforms LLM-as-a-judge baselines and matches or exceeds modern confidence-based methods in reranking candidates, improving both top-1 and majority-vote accuracy across AIME 24/25 and GPQA. As a highlight, on AIME 24 with a 1.5B model, CLUE boosts accuracy from 56.7% (majority@64) to 70.0% (top-maj@16).

## 1 INTRODUCTION

The remarkable ability of Large Language Models (LLMs) to generate numerous potential solutions for complex problems has also created a difficult new challenge: verification (Cobbe et al., 2021; Lightman et al., 2023; Hosseini et al., 2024). When a model produces dozens of different, plausible-looking answers for a single prompt, the task is no longer just about generation. It becomes a critical problem of selection: how can we reliably find the single correct answer within a flood of convincing but incorrect alternatives? This "selection problem" has become a major bottleneck, limiting the trust and real-world application of LLMs in high-stakes fields like mathematics and science, where correctness is absolutely essential.

To address this, the research community has largely pursued two main strategies. The first operates purely on the surface of the generated text, delegating evaluation to an external judge. This includes training separate reward models (Ouyang et al., 2022; Bai et al., 2022; Zheng et al., 2023) or adopting simple heuristics such as majority voting (Wang et al., 2022). While useful in practice, these post-hoc approaches are fundamentally blind to the model's actual reasoning process. They can be systematically misled by stylistic artifacts—for instance, verbose but incorrect answers often receive higher scores than terse but correct ones (Glaese et al., 2022). Moreover, trained judges inherit biases and limitations from their training data, making them brittle under distribution shift and expensive to retrain for new domains.

A second line of work attempts to go beneath the surface by relying on the model's reported confidence. Methods in this category use token probabilities, entropy, or derived uncertainty estimates (Kadavath et al., 2022b; Lin et al., 2023; Geng et al., 2023; Xiong et al., 2024; Fu et al., 2025b), assuming that higher probability correlates with correctness. However, LLM calibration remains a well-known

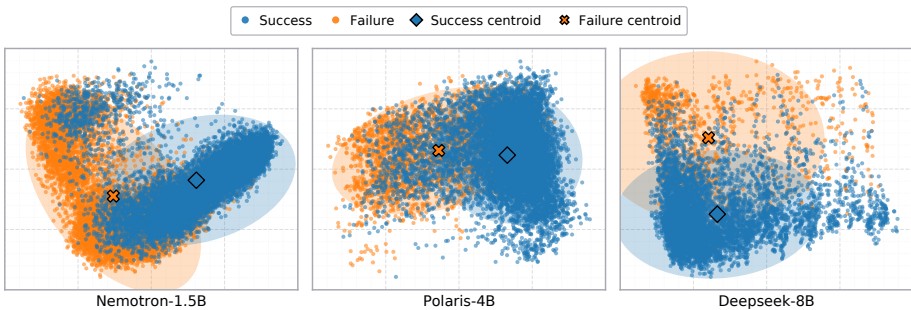

Figure 1: Visualization of activation deltas for correct (blue) and incorrect (orange) solutions from our experience set, projected to 2D using PCA. Each panel displays data from a different base model. Across all models, a distinct geometric separation between the two classes is visible.

weakness. Even state-of-the-art models are often "confidently wrong," assigning high likelihood to factually false or logically inconsistent outputs (Fu et al., 2025a). These confidence-based metrics degrade significantly on smaller or less-tuned models where probability distributions are noisier, making them a fragile basis for reliable verification.

In this work, we argue that the signal for correctness lies neither in the final generated text nor in token-level confidence alone, but rather in the geometry of the model's internal hidden states. Hidden states offer a unified representation, naturally subsuming both external and internal signals: early layers encode rich semantic features that inform text-based judgments, while later layers align more closely with the output logits that determine confidence. Our central hypothesis is that the process of arriving at a correct solution induces a characteristic and consistent transformation within the model's hidden space. To isolate this transformation, we focus on the **activation delta**—the difference between the hidden states at the beginning and end of the model's explicit reasoning process (i.e., between `<think>` and `</think>` tags). This delta intuitively factors out the initial prompt's influence and captures the geometric footprint of the computation itself. Crucially, this hypothesized structure is empirically observable. As shown in Figure 1, when we project these activation deltas into 2D space, a clear geometric separation emerges between correct and incorrect solutions. This clustering is not an isolated finding; it appears consistently across different models, suggesting a fundamental property of how these networks process information.

This visually evident structure motivates a deliberately lightweight verifier. If correct and incorrect reasoning trajectories produce such separable footprints, then an expensive, trained judge may be unnecessary. We introduce CLUE (**Clustering and Experience-based Verification**), a training-free framework that operates directly on these activation deltas. From a small set of labeled historical examples, CLUE computes two reference centroids: one for successful reasoning traces and one for failures. To classify a new solution, it simply computes its activation delta and identifies the closest centroid (using a layer-averaged Euclidean distance).

Our experiments demonstrate that this simple approach is remarkably effective. CLUE consistently matches or surpasses strong LLM-as-a-judge using GPT-4o and confidence-based baselines (Fu et al., 2025b), with especially clear gains on smaller or less-calibrated models where other signals are unreliable. Because CLUE is a deterministic, one-time computation without any gradient-based training, it is robust to the overfitting failures common in learned verifiers and generalizes well across diverse tasks and model scales. More broadly, our results provide strong evidence that the internal geometry of LLM hidden states contains a rich and reliable signal for verification.

## 2 THE CLUE FRAMEWORK

We first outline the core intuition behind CLUE. Each time an LLM solves a problem, its internal computation traces a trajectory through a high-dimensional representation space. We hypothesize that trajectories leading to correct solutions differ systematically from those leading to incorrect ones. CLUE captures this difference via a training-free, supervised aggregation over activation deltas. This section formalizes the setup and the resulting geometric rule.

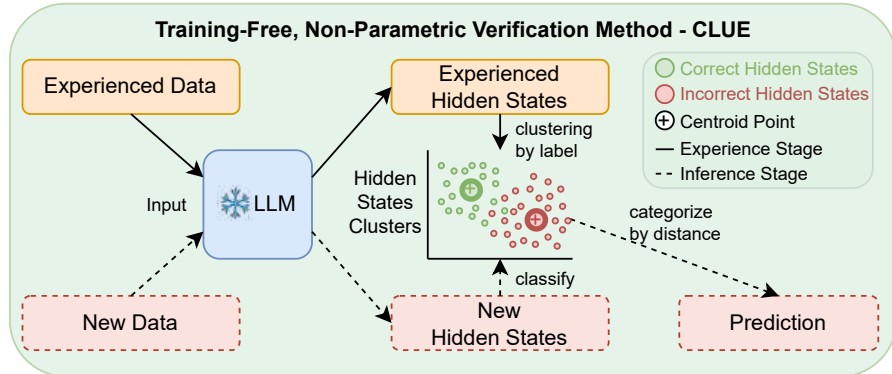

Figure 2: Overview of CLUE. **Left (Learning):** Labeled historical trajectories are summarized by their activation deltas and aggregated into success and failure centroid matrices. **Right (Verification):** A new trajectory is summarized by its activation delta and classified by the layer-averaged Euclidean distance (Eq. 3) to the two pre-computed centroids. There is no trained parameter throughout.

## 2.1 PROBLEM FORMULATION

Let an LLM be tasked with generating a response $R_i$ for a prompt $P$. We define a *trajectory* (or *experience*) $T_i = (P, R_i)$, paired with a ground-truth binary label $y_i \in \{0, 1\}$, where $y_i = 1$ denotes a correct solution (success) and $y_i = 0$ denotes an incorrect solution (failure). The goal is to learn a verification function $f$ that maps a new trajectory $T_{\text{new}}$ to a prediction $\hat{y}_{\text{new}} = f(T_{\text{new}}) \in \{0, 1\}$. Unlike text-based or probability-based approaches, which inspect the model's output, our function $f$ operates exclusively on the internal hidden-state representations generated during the production of $R_{\text{new}}$. This shifts the locus of verification from the surface of the text to the dynamics of the model's computation, while keeping the LLM parameters themselves frozen.

## 2.2 CLUE: VERIFICATION VIA ACTIVATION-DELTA SUMMARIES

Our central hypothesis is that the *transformation* of internal states during explicit reasoning contains a robust signal of correctness. To isolate this signal, we capture the transformation with an *activation delta*, defined as the difference between hidden states at the start and the end of the reasoning block. This approach is designed to factor out the initial prompt's conditioning and summarize the net effect of the reasoning computation itself. In our experiments, the reasoning block is consistently delimited by `<think>` and `</think>` tokens.

Let the model have $L$ layers and a hidden dimension $D$. For a given trajectory $T$, we denote by

$$\mathbf{h}_{\text{start}}(T) \in \mathbb{R}^{L \times D} \quad \text{and} \quad \mathbf{h}_{\text{end}}(T) \in \mathbb{R}^{L \times D}$$

the matrices of hidden states extracted, respectively, at the final token of `<think>` (just before reasoning) and at the final token of `</think>` (just after). The activation delta is the matrix

$$\Delta \mathbf{h}(T) = \mathbf{h}_{\text{end}}(T) - \mathbf{h}_{\text{start}}(T) \in \mathbb{R}^{L \times D}. \tag{1}$$

We use hidden states from *all* layers, reflecting the assumption that correctness-related information is distributed across the model's depth. Earlier layers retain rich semantic cues, while later layers align more strongly with the final output logits. The activation-delta matrix $\Delta \mathbf{h}(T)$ thus serves as a holistic feature representation for verification.

## 2.3 CENTROID CONSTRUCTION AND CLASSIFICATION

Our learning phase eschews gradient-based optimization in favor of a one-time, deterministic statistical aggregation over a labeled set of trajectories $\mathcal{D} = \{(T_i, y_i)\}_{i=1}^{N}$. First, we partition the dataset by outcome, defining index sets for success and failure:

$$\mathcal{I}_{\text{succ}} = \{ i \mid y_i = 1 \}, \qquad \mathcal{I}_{\text{fail}} = \{ i \mid y_i = 0 \}.$$

For each trajectory, we compute its activation delta $\Delta\mathbf{h}_i = \Delta\mathbf{h}(T_i)$ as in Eq. equation 1. The success and failure *centroid matrices* are then simply the element-wise means over each class, representing archetypal transformations for correct and incorrect reasoning:

$$\mathbf{V}_{\text{succ}} = \frac{1}{|\mathcal{I}_{\text{succ}}|} \sum_{i \in \mathcal{I}_{\text{succ}}} \Delta\mathbf{h}_i, \qquad \mathbf{V}_{\text{fail}} = \frac{1}{|\mathcal{I}_{\text{fail}}|} \sum_{i \in \mathcal{I}_{\text{fail}}} \Delta\mathbf{h}_i. \tag{2}$$

Both $\mathbf{V}_{\text{succ}}, \mathbf{V}_{\text{fail}} \in \mathbb{R}^{L \times D}$ are pre-computed and stored for inference.

At inference time, a new trajectory $T_{\text{new}}$ is summarized by its activation delta $\Delta\mathbf{h}_{\text{new}} = \Delta\mathbf{h}(T_{\text{new}})$. We classify it based on its proximity to the two reference centroids, using the *layer-averaged Euclidean distance*. For two matrices $\mathbf{A}, \mathbf{B} \in \mathbb{R}^{L \times D}$ with row vectors $\mathbf{a}_l, \mathbf{b}_l \in \mathbb{R}^D$, we define this distance as:

$$d(\mathbf{A}, \mathbf{B}) = \frac{1}{L} \sum_{l=1}^{L} \|\mathbf{a}_l - \mathbf{b}_l\|_2. \tag{3}$$

This metric gives equal weight to signals across all layers, preventing any single layer from dominating the classification. We compute distances $d_{\text{succ}} = d(\Delta\mathbf{h}_{\text{new}}, \mathbf{V}_{\text{succ}})$ and $d_{\text{fail}} = d(\Delta\mathbf{h}_{\text{new}}, \mathbf{V}_{\text{fail}})$, and classify via a nearest-centroid rule:

$$\hat{y}_{\text{new}} = \begin{cases} 1, & \text{if } d_{\text{succ}} < d_{\text{fail}}, \\ 0, & \text{otherwise}. \end{cases}$$

This simple rule, illustrated in Figure 2, requires no trainable parameters.

## 2.4 Application to Solution Reranking

The geometric formulation provides a continuous quality score that is naturally suited for reranking multiple candidate solutions. Given a prompt $P$ and $k$ generated responses $\{R_1, \ldots, R_k\}$, we form their corresponding trajectories $\{T_1, \ldots, T_k\}$ and compute their activation deltas $\{\Delta\mathbf{h}_1, \ldots, \Delta\mathbf{h}_k\}$. For each candidate $j$, we define a score based on its proximity to the success centroid:

$$s_j = d(\Delta\mathbf{h}_j, \mathbf{V}_{\text{succ}}), \qquad \text{(lower is better)}. \tag{4}$$

A lower score $s_j$ indicates that the internal reasoning process for solution $j$ is geometrically closer to the archetypal pattern of success. We can rank candidates in ascending order of their scores. This ranking can be used for direct top-1 selection or to enhance aggregation schemes.

## 2.5 Rationale for a Minimalist, Experience-Based Design

CLUE is intentionally minimalist to isolate the contribution of the representation itself. If a simple, training-free geometric rule over activation-delta summaries yields strong verification performance, this provides evidence that correctness signals are geometrically encoded and separable in activation space. By leveraging the geometry of *how* solutions are computed, CLUE offers a lightweight and broadly applicable path for verification that complements text-level and confidence-based signals.

# 3 Experiments

To rigorously evaluate the effectiveness of our non-parametric, hidden-state-based verifier, we designed a series of experiments targeting both in-domain mathematical reasoning and out-of-distribution general reasoning tasks. Our evaluation is structured around two primary objectives: first, to assess the raw classification accuracy of our method in distinguishing correct from incorrect solutions, and second, to measure its ability to improve overall reasoning performance by reranking multiple candidate solutions.

## 3.1 Datasets and Model Configuration

Our methodology relies on an "experience set" to establish the geometric reference points for successful and failed reasoning. For this purpose, we curated a comprehensive collection of mathematical

problems from two standard benchmarks: AIME (from 1983 to 2023) (Veeraboina, 2023) and the MATH (Hendrycks et al., 2021) dataset (specifically, problems of level 3 to 5). These datasets provide a diverse and challenging foundation for learning the characteristic activation patterns of mathematical reasoning. To test the performance and generalization of our approach, we use three distinct test sets. For in-domain evaluation, we use AIME 2024 and AIME 2025, which follow the same distribution as our experience data. To assess out-of-distribution (OOD) generalization, we evaluate on GPQA (Rein et al., 2024), a benchmark focused on graduate-level questions that demand complex, general reasoning abilities beyond the mathematical domain.

Our experiments cover a range of model sizes and architectures to ensure our findings are not specific to a single model's capabilities. We selected three distinct reasoning models: **Nemotron-Research-Reasoning-Qwen-1.5B** (Liu et al., 2025), a smaller yet capable model; **Polaris-4B** (An et al., 2025), a mid-sized model; and **DeepSeek-R1-0528-Qwen3-8B** (Guo et al., 2025a), a larger and more powerful model. To test the sensitivity of our method to the length and complexity of the reasoning trace, we conducted experiments with varied generation lengths for each model, specifically 16k, 32k, and 64k tokens. We use recommended inference setting including temperature, system prompt, from their Huggingface repository.

The process for constructing the experience set was as follows: for each problem in the AIME and MATH datasets, we sampled 32 unique solutions from the respective model. Each generated solution was then evaluated using a deterministic, rule-based verifier to obtain a ground-truth label of correct or incorrect. From this large pool of labeled solutions, we randomly selected 10,000 correct and 10,000 incorrect trajectories to form a balanced experience set. This set was used to compute the success and failure centroids as described in Section 2. For the evaluation phase, we generated 64 candidate solutions for each problem in our test sets.

## 3.2 EVALUATION SETUPS AND BASELINES

We evaluate our method, which we refer to as CLUE, across two distinct experimental setups.

The first setup frames the task as a binary classification problem to directly measure the verifier's accuracy. For each of the 64 sampled solutions on the test sets, our CLUE method predicts a label of correct or incorrect based on whether the solution's activation delta is closer to the success or failure centroid. The ground truth for this task is again determined by the rule-based verifier and we compare our method against strong baselines. Specifically, we use **GPT-4o** (Hurst et al., 2024) in an LLM-as-a-judge capacity. We evaluate the judge in two settings to control for the information they can access: one where the full solution, including the entire <think> block, is provided to the LLM judge, and another where only the part after the thinking process is provided. The former tests the judge's ability to evaluate the reasoning process, while the latter tests its ability to verify the result itself. In both cases, only LLM generated answer is provided to the judge without the reference ground truth answer.

The second setup evaluates the practical impact of our method on improving final reasoning accuracy through reranking. Here, for each test problem, we use CLUE to rerank the 64 generated solutions. The ranking criterion is the Euclidean distance of a solution's activation delta to the success centroid, with smaller distances indicating higher quality. We report our performance using several metrics: **top@1**, the accuracy of the single best-ranked solution; and **top-maj@k**, the accuracy achieved by performing majority voting on the answers from the top-$k$ ranked solutions, for $k \in \{4, 8, 16\}$. We compare these results against a suite of standard and state-of-the-art baselines. These include **mean@64**, which measures the average accuracy of a single randomly sampled solution; **majority@64**, the accuracy of standard majority voting over all 64 samples; **DeepConf@64** (Fu et al., 2025b), a recent and competitive method that uses model confidence scores for reranking; and **pass@64**, which represents the oracle upper bound, indicating whether at least one correct answer exists among the 64 samples.

## 3.3 CLASSIFICATION PERFORMANCE

Table 1 presents the performance of our verifier compared to a strong LLM-as-a-judge baseline (GPT-4o) on solutions generated by both a smaller model (Nemotron-1.5B) and a more capable one (Polaris-4B). We report overall accuracy, as well as the True Positive Rate (TPR), which measures the

Table 1: Binary classification performance of different verifiers on solutions generated by Nemotron-1.5B and Polaris-4B. Our method (CLUE) is compared against an LLM-as-a-judge baseline. We report overall Accuracy, True Positive Rate (TPR), and True Negative Rate (TNR).

| Verifier Method | Nemotron-1.5B Solutions | | | Polaris-4B Solutions | | |
|---|---|---|---|---|---|---|
| | Accuracy (%) | TPR (%) | TNR (%) | Accuracy (%) | TPR (%) | TNR (%) |
| *Test Set: AIME 2024* | | | | | | |
| CLUE (Ours) | **80.9** | 72.9 | 87.4 | **81.1** | 89.5 | 51.3 |
| GPT-4o (Answer Only) | 58.6 | 57.2 | 59.7 | 80.1 | 84.3 | 63.1 |
| GPT-4o (Full Trace) | 47.5 | 45.8 | 48.2 | 64.3 | 70.9 | 50.2 |
| *Test Set: AIME 2025* | | | | | | |
| CLUE (Ours) | **85.2** | 82.9 | 86.4 | **77.7** | 80.7 | 70.1 |
| GPT-4o (Answer Only) | 59.2 | 60.8 | 58.3 | 73.0 | 85.1 | 34.4 |
| GPT-4o (Full Trace) | 47.1 | 48.6 | 46.7 | 59.3 | 69.8 | 27.6 |

ability to correctly identify successful solutions, and the True Negative Rate (TNR), which measures the ability to correctly identify failed solutions.

The results clearly indicate that our CLUE verifier provides a substantial and consistent advantage over the LLM-as-a-judge baseline. A key observation is that the LLM judge exhibits a strong optimistic bias, frequently misclassifying incorrect solutions as correct. This is evident in its consistently low True Negative Rate, which drops to a mere 34.4% for Polaris-4B solutions on AIME 2025. This inherent optimism explains why the LLM judge's performance appears to improve significantly when evaluating solutions from a stronger base reasoner like Polaris-4B. A more capable reasoner produces a higher proportion of correct solutions, which the LLM judge identifies with reasonable accuracy (high TPR). Consequently, the judge's primary weakness—its failure to reliably identify incorrect answers—has a diminished impact on its overall accuracy score simply because there are fewer negative samples to misclassify. In contrast, our CLUE method demonstrates a more robust and balanced performance profile. It maintains a very high TNR (up to 87.4%) when evaluating the weaker Nemotron-1.5B model, making it highly effective at filtering out the larger volume of incorrect attempts. Simultaneously, it achieves a high TPR on outputs from the stronger Polaris-4B model (up to 89.5%), showing it is equally adept at recognizing valid reasoning. This balanced capability makes our approach a more universally effective verifier, providing significant benefits regardless of the base model's reasoning proficiency.

### 3.4 RERANKING FOR ENHANCED REASONING ACCURACY

Moving beyond binary classification, we evaluate CLUE as a reranking tool to improve reasoning accuracy. By scoring and reordering 64 candidate solutions per problem, CLUE consistently outperforms majority voting on both in-domain AIME and out-of-domain GPQA benchmarks (Table 2). For example, with Nemotron-1.5B on AIME 24, "top-maj@16" reaches 70.0% versus 56.7% for "majority@64." Even "top@1" often surpasses majority voting, showing the effectiveness of CLUE.

This advantage extends to general reasoning: on GPQA, Polaris-4B achieves 59.6% with CLUE versus 56.6% with majority voting. Such transfer demonstrates that the geometric separation of success and failure in hidden states reflects a fundamental property of math reasoning, not domain-specific patterns. Compared with the confidence-based baseline DeepConf, CLUE exhibits greater robustness. While both methods perform strongly on DeepSeek-8B, DeepConf collapses on weaker models, often below majority voting. CLUE, however, maintains its edge across all scales, leveraging internal reasoning signals that remain geometrically separable even when output confidences are poorly calibrated. This highlights CLUE's broad applicability, particularly for smaller models where confidence cues are unreliable.

### 3.5 GENERALIZATION AND THE INFLUENCE OF TRAINING PARADIGMS

We next examine CLUE's behavior across training paradigms and models. We find that the geometric separability of success and failure in hidden states depends strongly on training methodology—specifically, the contrast between Supervised Fine-Tuning (SFT) and Reinforcement Learning (RL).

Table 2: Reasoning accuracy on AIME and GPQA test sets after reranking 64 candidate solutions. Results are presented as percentages (%). mean@64 represents the average accuracy of a single sample, while pass@64 is the oracle upper bound.

| Metric | Nemotron-1.5B | | | Polaris-4B | | | DeepSeek-8B | | |
|---|---|---|---|---|---|---|---|---|---|
| | AIME 24 | AIME 25 | GPQA | AIME 24 | AIME 25 | GPQA | AIME 24 | AIME 25 | GPQA |
| *Baselines* | | | | | | | | | |
| mean@64 | 45.0 | 35.0 | 41.9 | 79.2 | 75.4 | 55.2 | 87.1 | 75.8 | 54.86 |
| majority@64 | 56.7 | 36.7 | 44.4 | 80.0 | 80.0 | 56.6 | 90.0 | 83.3 | 61.11 |
| DeepConf@64 | 56.7 | 30.0 | 40.2 | 80.0 | 73.3 | 55.7 | **93.3** | **86.7** | 62.12 |
| pass@64 (Oracle) | 76.7 | 63.3 | 83.8 | 86.7 | 90.0 | 88.4 | 93.3 | 93.3 | 94.85 |
| CLUE *Reranking (Ours)* | | | | | | | | | |
| top@1 | 66.7 | 40.0 | 46.5 | **83.3** | 76.7 | 52.5 | 90.0 | 83.3 | 56.57 |
| top-maj@4 | **70.0** | 40.0 | 43.9 | **83.3** | 76.7 | 57.1 | 90.0 | **86.7** | 61.62 |
| top-maj@8 | **70.0** | 40.0 | **47.0** | 80.0 | 80.0 | 58.1 | **93.3** | **86.7** | 61.11 |
| top-maj@16 | **70.0** | **43.3** | 44.4 | 80.0 | **83.3** | 59.6 | **93.3** | **86.7** | **62.63** |

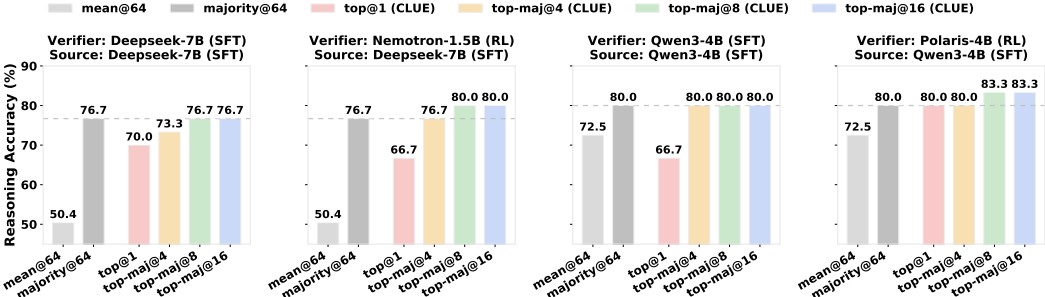

Figure 3: Cross-model reranking performance on AIME 24. The results show that RL-trained models (Nemotron-1.5B, Polaris-4B) are not only effective at self-verification but are also superior verifiers for trajectories generated by SFT-trained models (Deepseek-7B, Qwen3-4B).

We evaluated four models: two SFT/distillation-based (**Deepseek-7B** (Guo et al., 2025a), **Qwen3-4B** (Yang et al., 2025a)) and two RL-tuned (**Nemotron-1.5B**, **Polaris-4B**). In a cross-model setup, reasoning traces from one model were fed into another's hidden states for reranking, enabling both self- and cross-verification tests.

As shown in Figure 3, SFT models struggle: their self-reranking ("top-maj@16") barely matches or even lags the "majority@64" baseline, indicating weak internal separation of correctness. By contrast, RL models act as strong verifiers even across models: Nemotron-1.5B boosts Deepseek-7B's accuracy to 80.0% (vs 76.7% baseline), and Polaris-4B lifts Qwen3-4B's outputs to 83.3% (vs 80.0% self-rerank). We attribute this gap to training signals. SFT trains imitation of correct paths but lacks explicit negative feedback. RL, especially with verifiable rewards, supplies direct contrastive supervision, producing geometrically distinct clusters for correct vs. incorrect reasoning. This makes RL-trained models inherently stronger verifiers, both for themselves and others.

### 3.6 GENERALIZATION TO DIVERSE, NON-MATHEMATICAL REASONING

To test CLUE's generalization beyond mathematics, we evaluated it on the diverse **WebInstruct-verified** benchmark, which spans physics, law, finance, and the humanities. We collect experience to construct centroids from 5k training questions with generated solutions, and evaluation was conducted on 1k test questions. Ground-truth correctness labels were obtained by providing reference answers to GPT-4o, while GPT-4o itself—without access to the reference—served as the LLM-as-a-judge baseline.

As shown in Table 3, CLUE consistently outperforms GPT-4o across both 1.5B and 4B models. On the 1.5B model, CLUE reaches 60.4% accuracy versus GPT-4o's 54.0%. Most notably, for the 4B model, the LLM judge collapses to 48.1% (below random), while CLUE maintains 59.2%. These results

Table 3: Binary classification performance on the general-purpose WebInstruct-verified (Ma et al., 2025) dataset. We compare CLUE's accuracy against a GPT-4o judge on solutions generated by 1.5B and 4B models. The centroids for CLUE were computed using a subset of the training split.

| | Reasoner Model | |
|---|---|---|
| **Verifier Method** | **Nemotron-1.5B** | **Polaris-4B** |
| Test Set Composition (Success / Failure) | 1,263 / 2,737 | 1,584 / 2,024 |
| **CLUE (Ours)** | **60.4%** | **59.2%** |
| GPT-4o (LLM-as-a-judge) | 54.0% | 48.1% |

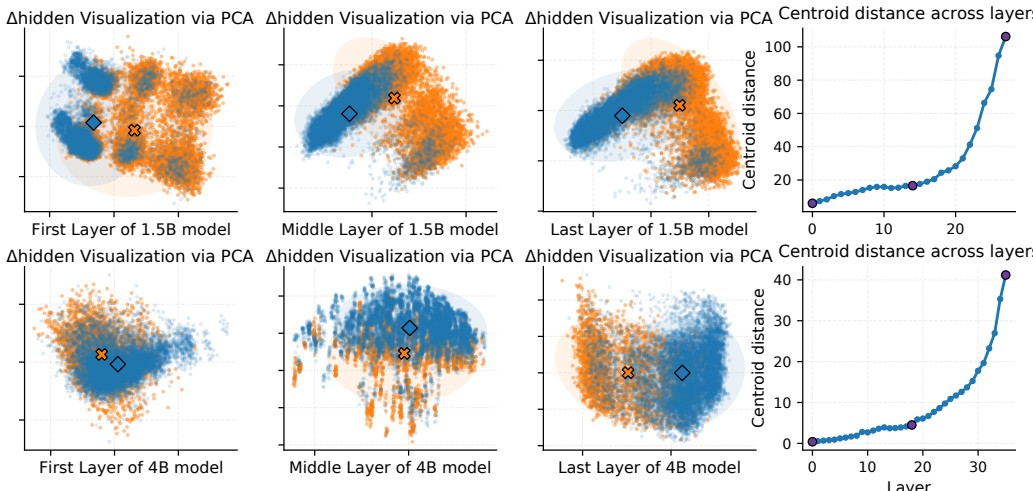

Figure 4: Layer-wise separability. Each row shows PCA projections from a shallow, a middle, and the final layer, plus a curve of the centroid distance $d^{(\ell)}$ across all layers. The centroid-distance curve increases with $\ell$, indicating stronger correctness signals at deeper layers.

provide strong evidence that correctness signals are encoded geometrically in hidden-state trajectories even outside mathematics. By collecting experience on a very small subset of training data, CLUE can extract a more stable and transferable representation of success versus failure, underscoring its robustness as a general-purpose verifier.

### 3.7 LAYER-WISE SEPARABILITY ANALYSIS

Next, we analyze the layer-wise structure of activation-delta matrices to visualize and quantify how class separability emerges from shallow to deep layers.

**Visualization.** We project layer-specific activation deltas onto two principal components via PCA. For a trajectory $i$ and layer $\ell$, let $\Delta\mathbf{h}_i^{(\ell)} \in \mathbb{R}^D$ denote the $\ell$-th row of $\Delta\mathbf{h}_i \in \mathbb{R}^{L \times D}$. We select representative shallow, middle, and final layers, apply PCA to $\{\Delta\mathbf{h}_i^{(\ell)}\}$, and plot the resulting 2D projections for successes and failures. As shown in the first three columns of each row in Figure 4, the classes are largely overlapping in shallow layers, begin to separate in middle layers, and form compact, well-defined clusters in the final layers.

**Quantification.** Let $\mathcal{I}_{\text{succ}}$ and $\mathcal{I}_{\text{fail}}$ be the index sets defined in §2.3. For each layer $\ell \in \{1, \ldots, L\}$, we compute layer-wise centroid by averaging the corresponding rows of the activation-delta matrices:

$$\mathbf{V}_{\text{succ}}^{(\ell)} = \frac{1}{|\mathcal{I}_{\text{succ}}|} \sum_{i \in \mathcal{I}_{\text{succ}}} \Delta\mathbf{h}_i^{(\ell)} \in \mathbb{R}^D, \qquad \mathbf{V}_{\text{fail}}^{(\ell)} = \frac{1}{|\mathcal{I}_{\text{fail}}|} \sum_{i \in \mathcal{I}_{\text{fail}}} \Delta\mathbf{h}_i^{(\ell)} \in \mathbb{R}^D. \tag{5}$$

We then measure the Euclidean distance between the two centroids at layer $\ell$:

$$d^{(\ell)} = \left\| \mathbf{V}_{\text{succ}}^{(\ell)} - \mathbf{V}_{\text{fail}}^{(\ell)} \right\|_2. \tag{6}$$

The rightmost panels of Figure 4 plot $d^{(\ell)}$ across layers. We observe a consistent upward trend, with the distance typically peaking in the final layers, aligning with the PCA visualizations and indicating that deeper representations encode more explicit and separable correctness signals.

# 4 RELATED WORK

## 4.1 LATENT REASONING AND ACTIVATION GEOMETRY

LLMs can reason in latent space instead of (or alongside) explicit token chains, via continuous "thought states" fed back into the model or compact hidden "thinking tokens" that compress CoT (Hao et al., 2024; Shen et al., 2025). Interpretability tools like the logit lens and tuned lens show that intermediate activations progressively align with output distributions, suggesting layer-wise decodable semantics and confidence signals (nostalgebraist, 2020; Belrose et al., 2023). Hidden-state probes can *self-verify* intermediate answers and enable early exit (Zhang et al., 2025), while semantic clustering of hidden rationales can improve robustness (Knappe et al., 2024). Beyond verification, activation directions can monitor or steer model traits (e.g., sycophancy, hallucination) via *persona vectors* (Chen et al., 2025). Also, in-context activation vectors indicate that linear structure in hidden space can be mapped and reused across tasks (Liu et al., 2024). More broadly, recent surveys on representation engineering highlight how linear directions and activation editing provide a general lens on hidden-state geometry in LLMs (Bartoszcze et al., 2025). Unlike these trained probes or steering methods, our verifier **CLUE** is training-free and purely reads cross-layer activation deltas.

## 4.2 TEST TIME SCALING

Recent research has increasingly focused on test-time scaling – techniques that improve model performance by allocating more computation during inference without changing the model's parameters. Parallel approaches (such as self-consistency (Wang et al., 2022) and ensemble "best-of-N" selection (Snell et al., 2024)) generate multiple independent chain-of-thought solutions and then aggregate or vote on the final answer, significantly boosting accuracy on complex tasks. Sequential approaches (such as iterative self-refinement (Madaan et al., 2023), Tree-of-Thoughts search (Yao et al., 2023)) allow the model to think in multiple steps, using intermediate reasoning to inform subsequent generations. Variants like weighted or semantic self-consistency (Luo et al., 2024; Knappe et al., 2024) highlight the importance of aggregating diverse rationales, while RLHF and LLM-as-a-judge approaches (Ouyang et al., 2022; Zheng et al., 2023) provide external supervision but can be costly and biased. To reduce dependence on large reward models, SWIFT learns *lightweight* hidden-state rewards that scale efficiently to best-of-$N$ sampling (Guo et al., 2025b). Complementary to this, DeepConf filters low-quality reasoning traces using internal confidence signals, improving both efficiency and accuracy (Fu et al., 2025b); relatedly, early-exit schemes can truncate overthinking while preserving accuracy (Kadavath et al., 2022a; Yang et al., 2025b). In contrast, **CLUE** introduces no trainable verifier: it computes success/failure centroids once from past experience and reranks by nearest centroid, showing that correctness is geometrically separable in hidden space.

# 5 CONCLUSION

In this work, we have demonstrated that the internal reasoning process of an LLM is not an inscrutable black box but a geometrically structured space containing clear, accessible signals of correctness. Our non-parametric framework, CLUE, validates this principle by achieving remarkable verification performance through simple geometric clustering of past experiences, proving more robust than both LLM-judges and confidence-based methods. Critically, we uncover a fundamental connection between a model's training paradigm and its internal geometry: models fine-tuned with Reinforcement Learning develop cleanly separable representations for correct and incorrect reasoning, a property largely absent in their SFT- counterparts. This insight suggests a paradigm shift for the field, moving beyond the evaluation of final outputs towards the direct analysis and shaping of the reasoning process itself. We believe that the success of our minimalist approach opens the door to developing a new class of lightweight, generalizable verifiers and inspires novel training objectives that explicitly optimize for internal representational clarity.

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

# A  APPENDIX

## A.1  ALGORITHMIC DETAILS

For completeness, we provide pseudocode for the two phases of CLUE: the one-time centroid aggregation (Algorithm 1) and the inference-time verification (Algorithm 2). Both phases operate on *activation-delta matrices* (§2.2) and use the *layer-averaged Euclidean distance* in Eq. equation 3. The underlying LLM remains frozen throughout.

Algorithm 1 constructs the reference centroids from a labeled set of trajectories. We first partition the dataset by ground-truth labels. For each trajectory, we extract the hidden-state matrices at the boundaries of the explicit reasoning block (<think> and </think>) and compute the activation delta $\Delta \mathbf{h}_i \in \mathbb{R}^{L \times D}$ via Eq. equation 1. We then compute the element-wise mean within each class to obtain the success and failure *centroid matrices* $\mathbf{V}_{\text{succ}}$ and $\mathbf{V}_{\text{fail}}$ (Eq. equation 2), which serve as geometric references during inference.

---

**Algorithm 1** Constructing CLUE Centroids (Learning Phase)

---

**Require:** Labeled dataset $\mathcal{D} = \{(T_i, y_i)\}_{i=1}^{N}$
 1: Initialize empty lists $\mathcal{H}_{\text{succ}}$, $\mathcal{H}_{\text{fail}}$
 2: Define index sets $\mathcal{I}_{\text{succ}} = \{ i \mid y_i = 1 \}, \mathcal{I}_{\text{fail}} = \{ i \mid y_i = 0 \}$
 3: **for** each $i \in \mathcal{I}_{\text{succ}}$ **do**
 4:     Extract $\mathbf{h}_{\text{start},i} \in \mathbb{R}^{L \times D}$ and $\mathbf{h}_{\text{end},i} \in \mathbb{R}^{L \times D}$
 5:     Compute $\Delta \mathbf{h}_i \leftarrow \mathbf{h}_{\text{end},i} - \mathbf{h}_{\text{start},i}$     (Eq. 1)
 6:     Append $\Delta \mathbf{h}_i$ to $\mathcal{H}_{\text{succ}}$
 7: **end for**
 8: **for** each $i \in \mathcal{I}_{\text{fail}}$ **do**
 9:     Extract $\mathbf{h}_{\text{start},i}$ and $\mathbf{h}_{\text{end},i}$
10:     Compute $\Delta \mathbf{h}_i \leftarrow \mathbf{h}_{\text{end},i} - \mathbf{h}_{\text{start},i}$
11:     Append $\Delta \mathbf{h}_i$ to $\mathcal{H}_{\text{fail}}$
12: **end for**
13: $\mathbf{V}_{\text{succ}} \leftarrow \text{mean}(\mathcal{H}_{\text{succ}})$     (Eq. 2)
14: $\mathbf{V}_{\text{fail}} \leftarrow \text{mean}(\mathcal{H}_{\text{fail}})$     (Eq. 2)
15: **return** $\mathbf{V}_{\text{succ}}, \mathbf{V}_{\text{fail}}$

---

Algorithm 2 describes the inference procedure. Given a new trajectory $T_{\text{new}}$, we compute its activation delta $\Delta\mathbf{h}_{\text{new}}$ as in Eq. equation 1, measure its distances to the two centroid matrices using Eq. equation 3, and decide by nearest centroid.

---

**Algorithm 2** Verification with CLUE (Inference Phase)

---

**Require:** New trajectory $T_{\text{new}}$; centroids $\mathbf{V}_{\text{succ}}$, $\mathbf{V}_{\text{fail}}$

1: Extract $\mathbf{h}_{\text{start,new}}$ and $\mathbf{h}_{\text{end,new}}$ from $T_{\text{new}}$
2: Compute $\Delta\mathbf{h}_{\text{new}} \leftarrow \mathbf{h}_{\text{end,new}} - \mathbf{h}_{\text{start,new}}$                                     (Eq. 1)
3: $d_{\text{succ}} \leftarrow d(\Delta\mathbf{h}_{\text{new}}, \mathbf{V}_{\text{succ}})$                                               (Eq. 3)
4: $d_{\text{fail}} \leftarrow d(\Delta\mathbf{h}_{\text{new}}, \mathbf{V}_{\text{fail}})$                                              (Eq. 3)
5: **if** $d_{\text{succ}} < d_{\text{fail}}$ **then**
6:     **return** 1                                                      ▷ classified as correct
7: **else**
8:     **return** 0                                                    ▷ classified as incorrect
9: **end if**

---

### A.2 ABLATION STUDIES

To validate the specific design choices of our CLUE methodology, we conducted a series of ablation studies. Our goal was to isolate the contributions of two key components: 1) the use of hidden states from all layers of the model, and 2) the computation of an activation *delta* ($\mathbf{h}_{\text{end}} - \mathbf{h}_{\text{start}}$) rather than using an absolute state vector. We evaluated three alternative configurations against our full method in Figure 4:

- **First Layer Only**: Using only the hidden states from the first transformer layer.
- **Last Layer Only**: Using only the hidden states from the final transformer layer.
- **Final State Only**: Using only the absolute hidden state vector at the end of the reasoning block ($\mathbf{h}_{\text{end}}$), without subtracting the baseline state.

The results of our ablation study confirm that each component of the CLUE verifier contributes to its overall effectiveness. The most significant finding is the critical role of network depth. Using only the **First Layer Only** leads to a dramatic performance collapse across all settings, indicating that the shallow, near-embedding layers of the model do not contain sufficiently abstract or separable representations to distinguish correct from incorrect reasoning. Conversely, the **Last Layer Only** variant performs remarkably well, achieving accuracy that is only marginally lower than our full method and even matching it in some cases. This demonstrates that the deepest layers of the model are the primary locus of the high-level reasoning signals we are leveraging. Finally, the **Final State Only** experiment highlights the benefit of our delta computation. While still a strong performer, removing the subtraction of the baseline state results in a noticeable performance degradation compared to the full CLUE approach.

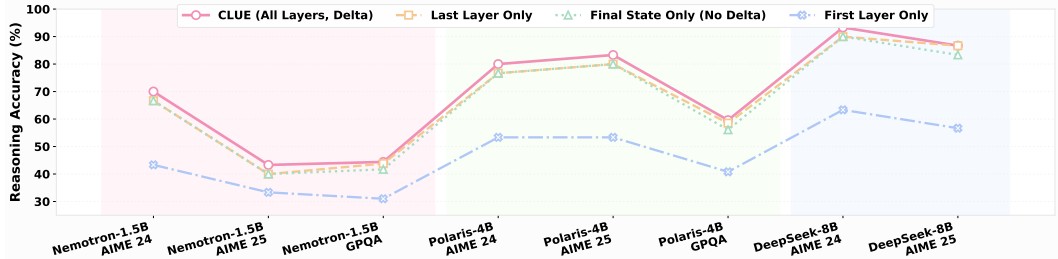

Figure 5: Ablation study results showing the top-maj@16 reasoning accuracy across different model and dataset configurations.

In summary, these ablations validate our methodology: the discriminative signal is strongest in deep layers, but leveraging the full stack of layers and calculating the activation delta are both important for achieving optimal performance.

### A.3 ANALYSIS OF RERANKING SCORE CALCULATION

In our reranking methodology, we employ a score based solely on the distance to the success centroid ($d_{\text{succ}}$), as defined in Equation 4. A natural question arises: why not use a comparative score that incorporates distance to both centroids, such as the difference $d_{\text{succ}} - d_{\text{fail}}$? Our choice is based on the hypothesis that the geometric cluster of successful reasoning trajectories is significantly more concentrated and consistent than that of failed trajectories. The path to a correct solution often follows a constrained logical sequence, resulting in a tight, well-defined cluster in the activation space. Conversely, errors can manifest in countless ways—from minor calculation mistakes to fundamental logical fallacies—leading to a "failure" cluster that is inherently more diffuse and scattered.

To validate this hypothesis, we conducted an experiment comparing our standard reranking method (using only $d_{\text{succ}}$) against an alternative that reranks candidates based on the score $d_{\text{succ}} - d_{\text{fail}}$. We evaluated both approaches on the AIME 24 and AIME 25 datasets using solutions generated by Nemotron-1.5B and Polaris-4B.

The results, summarized in Table 4, show that the simpler $d_{\text{succ}}$ metric is a more robust and generally higher-performing choice. For the Nemotron-1.5B model, using only the success centroid distance provides a clear advantage. On AIME 24, it achieves up to 70.0% accuracy, while the comparative score peaks at 66.7%. For the more capable Polaris-4B model, the two methods perform very similarly, often identically. However, the simpler $d_{\text{succ}}$ score still delivere a more consistent and strong performance. This evidence supports our initial intuition: the success centroid provides a stable and reliable signal. Incorporating distance from the more diffuse failure centroid does not consistently add value and can degrade performance.

Table 4: Comparison of reranking performance using two different scoring metrics. The simpler metric based solely on distance to the success centroid ($d_{\text{succ}}$) shows a clear advantage, especially for top@1 selection. All data for our method ($d_{\text{succ}}$) is consistent with the main results in Table 2. Results are reported as accuracy (%).

| Dataset | Reranking Metric | Nemotron-1.5B | | | | Polaris-4B | | | |
|---|---|---|---|---|---|---|---|---|---|
| | | top@1 | top-maj@4 | top-maj@8 | top-maj@16 | top@1 | top-maj@4 | top-maj@8 | top-maj@16 |
| AIME 24 | $d_{\text{succ}}$ (Ours) | **66.7** | **70.0** | **70.0** | **70.0** | **83.3** | **83.3** | 80.0 | 80.0 |
| | $d_{\text{succ}} - d_{\text{fail}}$ | 60.0 | 66.7 | 66.7 | 66.7 | 80.0 | **83.3** | 80.0 | 80.0 |
| AIME 25 | $d_{\text{succ}}$ (Ours) | **40.0** | **40.0** | **40.0** | **43.3** | **76.7** | **76.7** | 80.0 | 83.3 |
| | $d_{\text{succ}} - d_{\text{fail}}$ | 36.7 | **40.0** | **40.0** | 40.0 | 73.3 | **76.7** | 80.0 | 83.3 |

## B  USE OF LARGE LANGUAGE MODELS IN PREPARATION

During the preparation of this manuscript, we made use of large language models (LLMs) as writing and programming assistants. The models supported us by refining phrasing, enhancing clarity, and ensuring grammatical accuracy in the text. They also assisted with generating boilerplate code, debugging, and structuring code for the experiments. Importantly, all model outputs—both text and code—were critically reviewed, revised, and validated by the authors. The final responsibility for the accuracy and appropriateness of the manuscript rests entirely with the authors.

