# OpenReview forum: "CLUE: Non-parametric Verification from Experience via Hidden-State Clustering"
_ICLR.cc/2026/Conference — ICLR 2026 Conference Withdrawn Submission_

### Official Review · Reviewer_Pp7g · 2025-10-18

**Soundness:** 2
**Presentation:** 3
**Contribution:** 2
**Rating:** 2
**Confidence:** 4

**Summary:**

The paper claims that after encoded into hidden states, reasoning traces that lead to correct answers are geometrically distinct from reasoning traces that lead to incorrect answers. Such geometric difference can be used to directly judge whether the reasoning trace is correct or not.

**Strengths:**

1. The paper is well presented.

**Weaknesses:**

Baselines are very week. There are many SOTA methods for LLM-as-a-judge. In the paper, the author only use GPT 4O as baseline for LLM-as-a-judge and doesn't share the prompt. Since the paper claims a better method for LLM output verification, LLM-as-judge baseline should be as strong as possible.

No fine-grained analysis of the hidden state delta. If this hidden state delta does contain significant information for the correctness of the reasoning chain, this delta can be leveraged for many uses, e.g. filtering high quality training data. I suggest authors to conduct more experiments on proving the use of the hidden state delta, if it indeed has information about the correctness of the reasoning trace.

**Questions:**

N/A.

---

### Official Review · Reviewer_anmC · 2025-10-30

**Soundness:** 2
**Presentation:** 2
**Contribution:** 3
**Rating:** 4
**Confidence:** 4

**Summary:**

This paper proposes CLUE, a non-parametric reasoning verifier that estimates the correctness of reasoning traces via nearest-neighbor distances in embedding space.
The approach avoids fine-tuning by leveraging annotated reasoning traces and a KNN-based scoring function to verify whether a reasoning step is consistent with known correct examples.
Experiments are conducted to show better F1 than existing verifiers like SelfCheckGPT and JudgeLM.

**Strengths:**

- **Interesting Concept:** The idea of non-parametric verification via embedding neighborhoods is creative and simple.

- **Training-Free Nature:** Avoids fine-tuning large verifiers, making it lightweight and flexible.

- **Empirical Improvement:** Shows consistent gains across reasoning datasets.

**Weaknesses:**

- Limited Scope of Evaluation

The paper claims a general-purpose verifier, but experiments are limited to only 2–3 datasets.
For such a large claim, broader testing (e.g., code reasoning, multi-hop QA, or math proofs) would be needed.

- Lack of Mechanistic Explanation

The method’s working principle—why neighborhood density corresponds to reasoning correctness—is not theoretically or empirically justified.
The paper merely shows it “works” without explaining why.

- Ambiguous Claims of Scope

The title and abstract emphasize “non-parametric verification” as if generally applicable, but the method implicitly assumes reasoning trace annotations, which narrows its applicability.

- Baseline Incompleteness

The proposed method uses a large annotated trace set for KNN scoring, but fails to compare against fine-tuned model judgers, which could perform comparably or better.
The paper should include a baseline where the judging model is fine-tuned on the same data.

**Questions:**

Please refer to the weaknesses

---

### Official Review · Reviewer_5aDZ · 2025-11-01

**Soundness:** 1
**Presentation:** 3
**Contribution:** 1
**Rating:** 2
**Confidence:** 5

**Summary:**

The paper claims that there is a "geometrically structured space containing clear, accessible signals of correctness" in the LLM's "internal hidden states." These "internal hidden states" are the internal activation vectors between \<think\> and \</think\> delimiters. For a given trajectory, they define $h_{start}(T)$ and $h_{end}(T)$. The difference between these two matrices is called the activation delta, $\Delta h(T)$. Given a labeled dataset with {correct and incorrect} class labels, they compute element-wise means over each class. The resulting matrices are the "success and failure" centroid matrices. At inference time, they measure the distance of a new trajectory to the centroids and label it based on its closest centered. They also provide an extension to ranking when one has multiple answers to a prompt.

**Strengths:**

The method is simple and efficient.

**Weaknesses:**

- No theoretical explanations are provided for why the separability in the internal hidden states exists.

- The method requires clear correct and incorrect labels.

- The authors should choose Matthews Correlation Coefficient (MCC; https://en.wikipedia.org/wiki/Phi_coefficient) for their binary classification results.

**Questions:**

- Are there theoretical justifications or explanations for why the separability in the internal hidden states exists?

- How does this method do on general prompts where factuality is blurry?

---

### Official Review · Reviewer_1RxZ · 2025-11-02

**Soundness:** 3
**Presentation:** 3
**Contribution:** 2
**Rating:** 4
**Confidence:** 3

**Summary:**

The paper presents CLUE (Clustering and Experience-based Verification), a deliberately minimalist and non-parametric verifier for large language models (LLMs). Instead of training an additional verifier model, CLUE leverages the geometric properties of hidden-state activations to distinguish between correct and incorrect reasoning traces. Empirically, CLUE consistently outperforms LLM-as-a-judge baselines and matches or exceeds modern confidence-based reranking methods.

**Strengths:**

- The paper proposes a simple yet effective approach to verification that requires no additional training
- CLUE demonstrates strong empirical performance across multiple datasets, validating the usefulness of the proposed method

**Weaknesses:**

- Figure 1 does not seem to have a distinct geometric separation. There appear to be many overlapping regions across different models, which somewhat weakens the claim of clear separability.
- The paper lacks discussion on the effect of different hyperparameter choices related to the experience data, such as the size of the experience set, the number of rollouts per prompt, etc.
- For the classification performance in Section 3.3, the paper should include results on out-of-distribution (OOD) data to better demonstrate the generalizability of CLUE as a verifier.
- Based on the experiments in Section 3.7, it is unclear why the proposed method uses all layers for computing activation-delta matrices instead of focusing on the final layer, which appears to provide stronger separability. The paper also lacks ablation studies evaluating the design choice of using different layers of hidden states.

**Questions:**

- Can CLUE generalize effectively to open-ended tasks?

---

### Note · Authors · 2026-01-02

I have read and agree with the venue's withdrawal policy on behalf of myself and my co-authors.